# Neurological Evaluation of Patients with Newly Diagnosed Coeliac Disease Presenting to Gastroenterologists: A 7-Year Follow-Up Study

**DOI:** 10.3390/nu13061846

**Published:** 2021-05-28

**Authors:** Marios Hadjivassiliou, Iain D. Croall, Richard A. Grünewald, Nick Trott, David S. Sanders, Nigel Hoggard

**Affiliations:** 1Academic Department of Neurosciences, Sheffield Teaching Hospitals NHS Trust, Royal Hallamshire Hospital, Glossop Road, Sheffield S10 2JF, UK; r.a.grunewald@sheffield.ac.uk; 2Department of Infection, Immunity & Cardiovascular Disease, University of Sheffield, Sheffield S10 2JF, UK; i.croall@sheffield.ac.uk (I.D.C.); n.hoggard@sheffield.ac.uk (N.H.); 3Institute for in Silico Medicine, University of Sheffield, Sheffield S10 2JF, UK; 4Department of Dietetics, Sheffield Teaching Hospitals NHS Trust, Royal Hallamshire Hospital, Glossop Road, Sheffield S10 2JF, UK; nick.trott@nhs.net; 5Academic Department of Gastroenterology, Sheffield Teaching Hospitals NHS Trust, Royal Hallamshire Hospital, Glossop Road, Sheffield S10 2JF, UK; david.sanders1@nhs.net

**Keywords:** coeliac disease, neurological dysfunction, ataxia, headaches, neuropathy, anti-gliadin antibodies, MR imaging, TG6 antibodies

## Abstract

We have previously shown that 67% of patients with newly diagnosed coeliac disease (CD) presenting to gastroenterologists have evidence of neurological dysfunction. This manifested with headache and loss of co-ordination. Furthermore 60% of these patients had abnormal brain imaging. In this follow-up study, we re-examined and re-scanned 30 patients from the original cohort of 100, seven years later. There was significant reduction in the prevalence of headaches (47% to 20%) but an increase in the prevalence of incoordination (27% to 47%). Although those patients with coordination problems at baseline reported improvement on the gluten free diet (GFD), there were 7 patients reporting incoordination not present at baseline. All 7 patients had positive serology for one or more gluten-sensitivity related antibodies at follow-up. In total, 50% of the whole follow-up cohort were positive for one or more gluten-related antibodies. A comparison between the baseline and follow-up brain imaging showed a greater rate of cerebellar grey matter atrophy in the antibody positive group compared to the antibody negative group. Patients with CD who do not adhere to a strict GFD and are serological positive are at risk of developing ataxia, and have a significantly higher rate of cerebellar atrophy when compared to patients with negative serology. This highlights the importance of regular review and close monitoring.

## 1. Introduction

Coeliac disease (CD) is an autoimmune disorder triggered by the ingestion of gluten in genetically susceptible individuals and it affects 1% of the population [1]. CD belongs to the spectrum of gluten-related disorders that encompass diverse manifestations, including dermatitis herpetiformis (DH) and neurological dysfunction (gluten ataxia, gluten neuropathy, gluten encephalopathy) [2].

To what extent do patients with classic CD presentation suffer from neurological problems is an ongoing enquiry. Research has found evidence of cognitive deficits [3], which may recover by a degree after starting the gluten-free diet (GFD) [4], which CD carries an increased risk of developing vascular dementia [5], and some reports of brain atrophy and white matter lesions alongside neurological problems, which mirror those found in patients presenting neurologically with gluten ataxia/gluten encephalopathy [6,7]. While this body of research is sometimes complicated by CD cohorts including patients recruited from neurology departments (thus raising questions of generalisability to “typical”, gastrointestinal patients), more recently a validation study based on UK national population data bank has replicated some of these, including evidence of cognitive problems and brain white matter changes compared to controls [8].

With almost all of these experiments being single-timepoint, what remains most understudied is the longitudinal outlook of these patients. Importantly, it must be addressed if the GFD is as an effective treatment in stopping these central nervous system injuries from progressing further. Currently, one leading hypothesis for the pathogenesis of these problems focus on similar mechanisms to the gastrointestinal phenotype; which circulating autoantibodies against gluten products exhibit cross-reactivity with key body tissues and cause downstream harm [9]. Gliadin antibodies (AGA) have for example been demonstrated to exhibit reactivity with brain blood vessel structures [10], while AGA status has been correlated with depressive symptoms in healthy controls [11]. The presence of transglutaminase 6 (TG6) antibodies have also been implicated as potentially diagnostic/pathogenic in the development of gluten ataxia [12], with research demonstrating TG6 expression in cerebellar tissue (the primary site of injury in the condition) [13]. Transglutaminase 2 antibodies (TG2), a major diagnostic indicator for CD, have also been found deposited around brain vessel walls in patients with gluten ataxia [14]. These findings raise the question: does achieving negativity for all of these antibodies through the GFD [15] slow or even stop progression of neurological harm in CD patients?

In 2010 we conducted a 3-year prospective study with the primary aim of establishing the prevalence of neurological involvement at the time of diagnosis of CD in patients presenting to gastroenterology clinics with the typical gastroenterological symptoms of CD [16]. We also investigated any association between the presence of circulating TG6 antibodies and neurological deficits. The study demonstrated that out of 100 patients with newly diagnosed CD, 67% had symptoms and/or signs of neurological dysfunction. Sixty percent had abnormal brain imaging, including abnormal MR spectroscopy of the cerebellum in 46% and/or brain white matter lesions over and above what is expected from age in 25%. Forty percent of patients who had circulating TG6 autoantibodies displayed atrophy of subcortical regions on brain imaging, particularly involving the thalamus and the cerebellum, when compared to those patients with no TG6 antibodies.

In this follow-up study we reviewed 30 of the original participating patients 7 years after their baseline assessment. They underwent detailed clinical examination and repeat brain imaging. This investigation aimed to characterise any change in the clinical (neurological) phenotype of these patients, and also to investigate for relationships between antibody status and rate of brain atrophy.

## 2. Methods

### 2.1. Patient Selection and Clinical Assessments

The study was based at the Departments of Gastroenterology, Academic Department of Neurosciences and Neuroradiology, Sheffield Teaching Hospitals NHS Trust, Sheffield, UK. The study (research number STH20656) was approved by the local ethics committee and informed consent was obtained from all participants. All 100 patients who had participated in the original study were send an invitation to participate in the follow-up study. The ethics approval allowed for only a single invitation letter to be send to each of the 100 original participants. Thirty patients who responded to this invitation were seen and reassessed and underwent further brain imaging as per the original protocol (see below). Reasons for patients non-responding included: patients had moved away from Sheffield (the original cohort was from the Sheffield catchment area), and the COVID pandemic, which disrupted all non-COVID research in the UK for 9 months. Additionally, the terms of ethical approval did not allow for a second invitation letter to be sent at a later date. All responders were seen.

As per the original protocol, all patients had been diagnosed with CD following gastroscopy and duodenal biopsy [3]. All patients were clinically assessed by a consultant neurologist, including detailed neurological history and examination. The clinical examination included detailed assessment of gait, including ability to tandem walk, stand on each leg in turn and ability to stand with feet together. All patients had serological testing for gluten sensitivity during their attendance (see below).

### 2.2. Brain Imaging

Patients underwent MR imaging of the brain, using the same scanner (3T Philips Ingenia, AE Eindhoven, The Netherlands) and sequences as in the original study. Acquisitions included volumetric T1-weighted (T1W) and volumetric FLAIR structural imaging, and MR spectroscopy of the cerebellum (vermis). The methodology of the MR spectroscopy acquisition and the cut-offs for abnormal spectroscopic measurements NAA/Cr (N-Acetyl Aspartate to Creatine ratio) have been described previously. The T1W sequence was a 3D “MPRAGE” routine, 0.8 cm^3^ isotropic resolution, TR/TE = 10.5/4.8 ms. The FLAIR sequence was a 3D routine, 0.97 × 0.97 × 0.56 cm resolution, TR/TE = 4800/304 ms, TI = 1650 ms.

The original study had indicated that both cerebellar grey matter (GM) and the thalamus may be at risk of atrophy in this patient cohort. Accordingly, volumetry of these regions was performed on the newer scans and re-performed on the baseline scans, to ensure software/workstation comparability. All scans first underwent bias correction using the “N4ITK” package. For estimated of cerebellar GM the “SUIT” pipeline was used (http://www.diedrichsenlab.org/imaging/suit.htm (accessed on 25 March 2021), Figure 1), while thalamic volumes were calculated using FSL’s “FIRST” pipeline. Each of these methods were used to produce volumes for the region of interest (ROI). For FIRST this was calculated as the volume of all voxels identified as left/right thalamus in the final, boundary-corrected output. For SUIT this was calculated as the mean*volume of all voxels in the initial cerebellar GM tissue probability map. These “raw” volumes were then calculated as a percentage of one another, per-patient, to reflect what percentage brain volume change (pBVC) had occurred over time. This percentage was normalised with reference to the time which had passed between their two scan dates, to calculate the percentage yearly brain volume change (pYBVC). Finally, FSL’s “SIENA” package was also run for an estimation of overall brain volume change over time. This similarly produces a pBVC value, which was also normalised to create pYBVC.

### 2.3. Serology

Serological testing for IgA transglutaminase 2-TG2 (Phadia, Thermo Fisher Scientific, Uppsala, Sweden), anti-gliadin (AGA) IgG and IgA (Phadia, Thermo Fisher Scientific, Uppsala, Sweden), endomysium-EMA (Werfen, Warrington, UK), and IgG and IgA transglutaminase 6-TG6 (Zedira, Darmstadt, Germany) antibodies was undertaken in all patients during their visit for clinical assessment. All tests were performed at the NHS immunology lab as part of normal clinical care. The exact kits used differed in some instances compared to patient’s baseline assessments (both for AGA and TG6 antibodies).

### 2.4. Statistical Analysis

Baseline neurological symptoms of the returning sample were compared by chi-square analysis to the original whole sample to determine if they were significantly different as a subgroup in some manner. Clinical characteristics were then summarised descriptively in terms of how the rate of neurological symptoms has persisted over time. As the kits used to test for gluten antibodies changed in some instances between timepoints these were not explored individually. Instead, where relevant patients were classified based on if they were either negative for all tests, or positive for any test at follow-up.

For imaging analyses, descriptive statistics were produced to show the rate of brain atrophy over the whole cohort. Statistical analysis then focused on comparing the rate of change (pYBVC) between those patients who still had positivity of any gluten-related antibody at follow-up, compared to those in which all antibodies were within normal levels. Age was first compared between these groups to determine if it was significantly different and therefore should be included as a covariate in subsequent analyses or not. Groupwise testing then compared pYBVC of all cerebellar GM (SUIT), the thalamus (FIRST), and overall brain volume (SIENA) between antibody groups, after visual inspection of histograms was performed to decide if parametric or non-parametric models should be used.

## 3. Results

### 3.1. Clinical Characteristics

The baseline and follow-up clinical and serological characteristics of the 30 patients are summarised in Table 1. There were 18 female and 12 male patients. The mean age at baseline (on presentation and diagnosis of CD) was 47.8 (range 20–69) and at follow-up 55 (range 26–76).

Amongst these 30 patients, at the time of diagnosis of CD (baseline study), clinical history revealed that 14/30 (47%) complained of frequent and intractable headaches, 8/30 (27%) complained of balance problems and 3/30 (10%) of sensory symptoms. At the same time, on examination, 10/30 (33%) had gait ataxia, 3 of which also had nystagmus and one had myoclonic tremor. These figures were not significant (Chi square) to what was observed in the whole cohort of 100 patients from which these 30 patients were re-assessed. Further comparisons of baseline data between the returning cohort and other groupings were therefore not sought.

At follow-up (average 7 years later) the clinical history revealed that only 6/30 (20% from the original prevalence of 47%) complained of ongoing headaches and 4 of these 6 said that their headaches were much better since being on gluten-free diet. Fourteen patients (47%) mentioned gait instability. Seven of these patients had not reported any balance problems at the time of their baseline assessments. Of the remaining 7 patients who had reported balance problems at baseline and at follow-up, 3 said that their balance had improved on GFD, the other 4 stated that it was stable. Sensory symptoms were reported by 3/30 (10%) patients.

On examination at follow-up 17/30 (57% as opposed to 33% at baseline) of patients had mild gait ataxia (tandem walking difficulties). Four patients (13% as opposed to 0% at baseline) had loss of vibration sensation in their feet. Two patients had nystagmus (the same 2 that had nystagmus at baseline) and one had myoclonus (same patient as baseline).

All 7 patients who did not have ataxia at baseline assessment but developed mild gait ataxia at follow-up were positive for one or more serological tests for gluten sensitivity. The remaining 10 patients had ataxia from baseline. Of the 10 patients, 3 were found to have a neuropathy at follow-up neurophysiological assessment, 3 showed improvement of their ataxia on GFD, 3 remained the same and in one the ataxia had completely resolved.

Neurophysiological assessments in the 4 patients with evidence of reduced vibration sensation in their feet, at follow-up, showed a peripheral neuropathy in 3 (2 large fibre and 1 small fibre neuropathy). Therefore, the prevalence of neuropathy in this cohort was 10%.

### 3.2. Serological Tests

Serological tests at follow-up showed that 15 (50%) of patients were positive for one or more tests signifying gluten sensitivity. These included AGA (9 patients) and/or TG6 antibodies (10 patients). Amongst those patients positive for TG6 and/or AGA antibodies there were 6 (20% of total of 30) who were positive for TG2 antibodies and 5 (17% of total of 30) who were positive for EMA. The mean TG2 titre amongst the 30 patients at baseline was 207 U/mL and at follow-up 6 U/mL (normal <7 U/mL).

### 3.3. Brain Imaging

Brain imaging was performed in 29 patients; one patient failed to attend their appointment. MR spectroscopy from the vermis (NAA/Cr) significantly improved in 12 patients (40%), remained the same in 10 (33%) and got worse in 7 (23%). Of the 7 patients with worse spectroscopy, 5 had positive serology.

Brain volume analysis was performed on 28 patients, as one subject was excluded due to undergoing a clinical scan routine at their follow-up appointment such that the T1W acquisition was not comparable to their baseline. For the remaining 28 patients, average percentage yearly brain volume change (pYBVC) of the different ROIs was as follows: cerebellar GM = −0.53% (±0.42); thalamus = −0.49% (±0.38); whole brain = −0.16% (±0.15).

Of these 28 patients, 14 still had positivity of at least one gluten-related antibody at follow-up. Those who were still antibody positive were not significantly different in age (mean = 55.3 ± 10.2 years) to those who had become antibody negative (mean = 53.2 ± 11.6 years, independent *t*-test *p* = 0.612). Independent *t*-tests revealed the rate of atrophy of cerebellar GM (i.e., pYBVC) was significantly greater for the antibody-positive group (mean pYBVC = −0.73% ± 0.45) compared to the antibody-negative group (mean pYBVC = −0.32% ± 0.26, *p* = 0.007, Figure 2). However, pYBVC was not significantly different between antibody groups for either the thalamus (*p* = 0.641) or the whole brain (*p* = 0.486).

## 4. Discussion

We previously showed that neurological involvement in classic CD is common at the time of diagnosis but is often overlooked [16]. In this follow-up study involving 30 patients from the original cohort, we demonstrate a significant reduction of frequent intractable headaches from 47% to 20% after the introduction of GFD over a period of 7 years. By contrast, we found a significant increase in the symptoms of gait instability from 27% to 47%. This was also associated with increase in the prevalence of mild gait ataxia on clinical examination from 33% to 47%. In addition, 3 patients (10%) developed a peripheral neuropathy. Of interest is the fact that all of the patients with new onset of mild gait ataxia on examination, had detectable abnormalities in one or more serological tests for gluten sensitivity, suggesting suboptimal adherence to GFD [15]. In fact, overall, 50% of the 30 patients assessed 7 years later still had positive serology for one or more serological markers, although the level was significantly lower than baseline measurements.

We have previously showed that patients presenting with gluten ataxia who adhere to a strict GFD as indicated by the complete elimination of all antibodies, improve [17]. Those who do not go on the diet, worsen and those who go on the diet but are still serologically positive, also worsen, but at a slower pace. Such findings demonstrate that serological positivity may at least be a marker of gluten-sensitive patients who are at continued risk of neurological injury. Indeed, as CD is autoimmune in nature the presence of these antibodies in the blood may be the actual driver of neurological problems, as they react with key tissue sub-types and promote a chronic, harmful inflammatory state [18].

Here, our observations again demonstrate the relevance of serological markers in assessing patients with CD presenting to the gastroenterologists; those patients with CD who developed mild ataxia during this 7-year interval had positive serological markers for gluten sensitivity. As strict adherence to the GFD is thought to lead to elimination of related antibodies [15], this implies inadequate adherence to GFD. The fact that these patients had a very mild gait ataxia as opposed to the severe ataxia we often observe in those patients presenting with gluten ataxia, suggests that a GFD that may not be 100% strict, may nonetheless offer some protection. Patients who present with gluten ataxia to neurologists are also on average 10 years older than those presenting with CD with gastroenterological symptoms [16]. More prolonged exposure to gluten in comparison to patients who are diagnosed with CD 10 years earlier may have a role to play in the development of neurological symptoms.

Patients also underwent repeat brain MRI, and analysis of this provides evidence of the importance of the GFD in preventing progression of neurological outcomes on a physiological level. Here, the rate of brain volume loss was estimated for three regions of interest (ROI); the cerebellar grey matter, the thalamus, and the whole brain. While the thalamus and whole brain analyses did not reveal significant findings, a significantly higher rate of atrophy was found in the cerebellum for those patients who still had antibody positivity at follow-up. The cerebellum was included as a ROI given its known relevance in gluten-related disorders. It is, for example, the primary site of damage in gluten ataxia [2], while the baseline study of this cohort indicated that TG6 antibody positivity was associated with lower cerebellar grey matter volume [16]. The current study shows, for the first time, direct evidence that achieving serological negativity with a strict GFD in patients with CD is associated with an at-least reduced progression of brain atrophy. That this has been shown in a group of patients diagnosed via gastroenterology, further underlines the importance of this observation for all patients with CD.

It is potentially concerning that the two brain ROIs, which were selected because they had been shown to be at risk of atrophy in this cohort—the cerebellum and thalamus—each showed a rate of atrophy almost three times as fast as that of the whole brain. Although those with antibody negativity have some level of protection, a faster rate of atrophy in at-risk ROIs may therefore be persisting even in those on well controlled diets. While it has been shown that different regions of the brain shrink at different rates in heathy ageing [19], this nonetheless presents an important detail for further study as it may compel a need for additional treatment (e.g., immunosuppressive medication).

Fifty percent of the patients assessed in this study had positive serology for one or more serological marker for CD 7 years later. Such an immunological response is associated with slow but definite progression of the ataxia. These observations highlight the importance of regular clinical review with repeat serological testing, not just with EMA and TG2 antibodies but also with AGA and TG6 antibodies. Of interest is the fact that only 37% of these 30 patients were still under active gastroenterology follow-up.

Two of the patients who participated in this follow-up study were subsequently followed-up in the gluten neurology clinic because of the presence of ataxia and an abnormal brain scan, in combination with positive serology. After a review by the dietitian, a repeat scan and serological testing approximately one year later showed significant improvement on MR spectroscopy associated with clinical improvement. This suggests that there might still be a potential for neurological improvement even in the context of previous imperfect adherence to GFD, assuming that there is still some cerebellar reserve and no permanent atrophy.

Four patients developed symptoms of peripheral neuropathy at follow up and in 3 the neurophysiological assessment showed large fibre neuropathy in 2, and small fibre neuropathy in one. This is in keeping with previous observations that peripheral neuropathy is a late manifestation in patients diagnosed with classic CD [20].

This study has its limitations. Our intention was to try and review as many patients as possible from the original cohort. We only managed to review 30 patients, although we did recruit and assess all patients that contacted us following the invitation letter. The smaller number was a result of the interruption of the study by the COVID-19 pandemic and the fact that many patients had moved out of the Sheffield area. Furthermore, the ethics approval prevented us from attempting to contact the patients for a second time, something that we did consider doing after ease of travelling restrictions due to the pandemic. However, we believe that this cohort of 30 patients is a representative group as indicated by the fact that the neurological symptoms and signs were not significantly different to those observed in the baseline cohort of 100 patients, and age and sex of the cohort at baseline was similar to that of the original group.

Future lines of investigation should seek to replicate these findings with larger cohorts, and where possible, more deeply explore the relationship between neurological symptoms/brain imaging findings and relevant clinical markers, such as specific gluten-related antibodies, other markers of dietary adherence, and also indices of cognition and quality of life. Molecular analysis of regulatory factors and of genes relevant to processes of neuroinflammation, neurodegeneration, and autoimmunity would also be desirable to further identify risk factors and better understand the pathogenesis of these findings.

In conclusion, we showed here that whilst the prevalence of headache improves significantly amongst patients with CD presenting to gastroenterologists, there is evidence of development of mild gait ataxia in some patients who have positive serology for gluten antibodies 7 years later. A significantly higher rate of atrophy was found in the cerebellum of those patients who still had antibody positivity at follow-up. This highlights the importance of regular review and close monitoring using a battery of serological tests. Those patients who remain positive need further dietetic input and closer monitoring.

## Figures and Tables

**Figure 1 nutrients-13-01846-f001:**
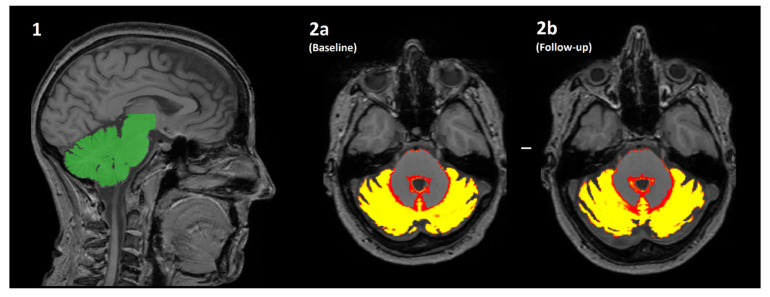
An overview of how the volume of cerebellar grey matter is estimated using the SUIT pipeline. (**1**) The cerebellum is automatically identified (green) and segmented from the original T1-weighted image. (**2a**/**2b**) The cerebellum is then further segmented into a “tissue probability map”, where the value of each pixel is transformed into the % probability that it is brain grey matter. The mean of all pixels with a non-zero value is multiplied by the volume of all such pixels (i.e., the mean*volume of red/yellow regions) for both the baseline (**2a**) and follow-up (**2b**) scans. The difference in values is the estimated change in volume over time, which is further normalised by the actual length of time between scanning sessions to produce percentage yearly brain volume change (pYBVC).

**Figure 2 nutrients-13-01846-f002:**
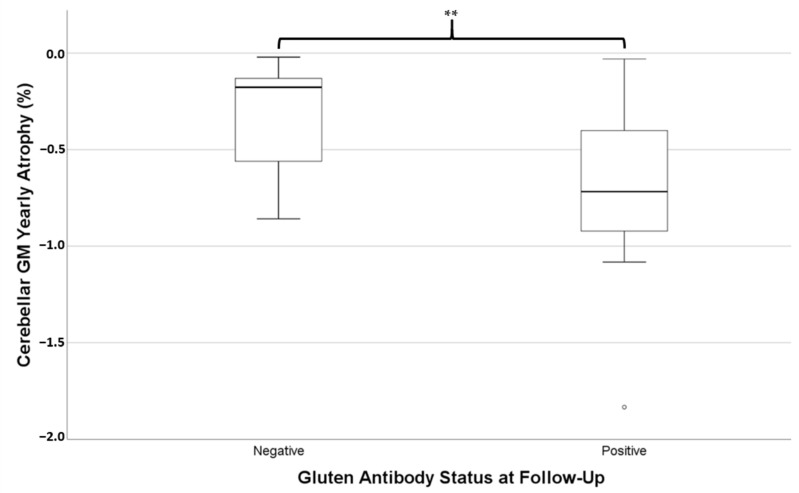
Percentage yearly brain volume change (pYBVC) of cerebellar grey matter, compared between patients who were still positive for gluten-related antibodies at follow-up and those who weren’t. A significantly higher rate of atrophy was found in the antibody-positive group (** *p* = 0.007).

**Table 1 nutrients-13-01846-t001:** Clinical and serological characteristics of the 30 patients that participated in the follow-up study both at baseline and at follow-up.

	Baseline (30 Patients)	Follow-Up 7 Years Later (Same 30 Patients)
mean age (range)	47.8 (20–69 years)	55 (26–76 years)
headaches	47%	20%
gait instability	27%	47%
sensory symptoms	10%	10%
gait ataxia on examination	33%	47%
sensory loss	0%	13%
EMA positive	93%	17%
anti-gliadin positive (IgG and/or IgA)	80%	30%
TG6 antibody positive (IgG and/or IgA)	37%	33%
TG2 IgA	100%	20%
one or more of the above serological tests positive	100%	50%
new onset of gait ataxia and positive serology at follow-up	N/A	7/7 100%

EMA, endomysium antibodies; TG6, transglutaminase 6; TG2, transglutaminase 2.

## Data Availability

Anonymised data can be provided on request.

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
