# Peer review of "Neurological Evaluation of Patients with Newly Diagnosed Coeliac Disease Presenting to Gastroenterologists: A 7-Year Follow-Up Study"

_nutrients, 2021, doi:10.3390/nu13061846_

Round 1

Reviewer 1 Report

General Comments:

Authors report some new data from the original clinical study involving 100 celiac patients that was conducted 7 years prior to the current/new study. All original 100 patients were examined for the presence of neurological dysfunction (67%) and were placed on gluten-free diet (GFD). Current/new study was conducted with select 30/100 original patients. Interestingly, after the 7 years of GFD, there were some measurable improvements in celiac serology, as well as in neurological function. These results and patients should be subject of further and more detailed examination.

Specific Comments:

  • Although there was no overall statistical difference in neurodegeneration (as evaluated by CNS imaging of brain atrophy) in select 30 CD patients at the time of diagnosis and 7 years later, in 12 of these patients, statistically significant improvement was observed. A figure (CNS-scan) that would illustrate such an improvement would strengthen the manuscript.
  • It would be of interest to determine if corresponding antibody decreases were also significant: It seems from the presented data in Table 1 that EMA, TG2 and AGA antibodies were significantly lower 7 years later while TG6 antibodies were not. These data could be shown more clearly in form of the (for example) Prism Dot Plots with indication of statistical level of significance.
  • Authors mention in the discussion that incomplete remission of some of the serological markers of CD could be due to incomplete adherence to GFD however, they do not offer any further corroboration or explanation. Please provide.
  • If not in this study, in future studies, it would be informative to perform detailed molecular analyses of potential regulatory (miRNA, mRNA and/or proteomics) factors and genes associated with neuroinflammation, neurodegeneration and autoimmunity using the biopsies (cerebral-spinal fluids or similar) or at least blood/GALT samples derived from these patients.
  • Authors should cite more than 6 references and include more information available in the field on this subject.

Author Response

We thank the reviewer for their time in assessing our manuscript. Below we outline a point by point response to each of the comments.

Authors report some new data from the original clinical study involving 100 celiac patients that was conducted 7 years prior to the current/new study. All original 100 patients were examined for the presence of neurological dysfunction (67%) and were placed on gluten-free diet (GFD). Current/new study was conducted with select 30/100 original patients. Interestingly, after the 7 years of GFD, there were some measurable improvements in celiac serology, as well as in neurological function. These results and patients should be subject of further and more detailed examination.

Specific Comments:

  • Although there was no overall statistical difference in neurodegeneration (as evaluated by CNS imaging of brain atrophy) in select 30 CD patients at the time of diagnosis and 7 years later, in 12 of these patients, statistically significant improvement was observed. A figure (CNS-scan) that would illustrate such an improvement would strengthen the manuscript.

Thank you for the suggestion. Our findings *do* report significant statistical difference with respect to neurodegeneration / brain atrophy over the 7 year period; in the imaging analysis a greater rate of atrophy was found in the cerebellum of the patients who remained antibody positive compared to those who had achieved antibody negativity. Presuming that this is what the reviewer is discussing in terms of a CNS-scan, the manner in which the analysis was conducted unfortunately does not straightforwardly translate into a figure for the manuscript. The image processing essentially used computer software to define the area of each scan which belonged to cerebellar grey matter, and then translate this into a number which represents its volume. These volumes were then compared in groupwise testing (t-tests), meaning that the boxplot already provided in the manuscript is arguably the most appropriate/conventional way of representing this data.

Understanding however that the image processing may be not be adequately conveyed in the initial draft, we have instead provided a new figure which illustrates some of these methods. The figure has been added into the methods rather than results so that it is not misinterpreted as being representative of any study findings.

  • It would be of interest to determine if corresponding antibody decreases were also significant: It seems from the presented data in Table 1 that EMA, TG2 and AGA antibodies were significantly lower 7 years later while TG6 antibodies were not. These data could be shown more clearly in form of the (for example) Prism Dot Plots with indication of statistical level of significance.

We agree with the reviewer that this is a very interesting point, however it is difficult to do in a controlled manner with the current dataset as the testing kits used for AGA and TG6 were different between baseline and follow-up. We believe this makes a statistical comparison of this data (i.e. positivity rates between different antibodies) inadvisable, and hope the reviewer understands. We have added into the main text about the kits changing over time so this point is clear for the reader (in the methods; under “Serology” and also under “Statistical Analysis”)

  • Authors mention in the discussion that incomplete remission of some of the serological markers of CD could be due to incomplete adherence to GFD however, they do not offer any further corroboration or explanation. Please provide.

A reference has been provided to support this point.

  • If not in this study, in future studies, it would be informative to perform detailed molecular analyses of potential regulatory (miRNA, mRNA and/or proteomics) factors and genes associated with neuroinflammation, neurodegeneration and autoimmunity using the biopsies (cerebral-spinal fluids or similar) or at least blood/GALT samples derived from these patients.

We agree with the reviewer these are extremely relevant points that should be explored. We have added a new paragraph to the discussion (the penultimate one) which comments on avenues for future research and includes these suggestions.

  • Authors should cite more than 6 references and include more information available in the field on this subject.

A number of references have been added throughout the introduction and discussion.

Reviewer 2 Report

The authors presented a follow-up analysis of a CD patients cohort after 7 years. They applied neuroimaging evaluation, as well as serum and clinical evaluation. The paper presents an interesting analysis but it has also several weaknesses that the authors might address:

  1. the introduction is too much focused on the previous work and does not present the state of the art of knowledge about the field. 
  2. There are results that have not been discussed from the perspective of the existing literature (e.g., immunological response).
  3. How is different the baseline evaluation of the responders from the non-responders? Have you looked for some predictors of drop-out

Author Response

We thank the reviewer for their time in assessing our manuscript. Below we outline a point by point response to each of the comments.

The authors presented a follow-up analysis of a CD patients cohort after 7 years. They applied neuroimaging evaluation, as well as serum and clinical evaluation. The paper presents an interesting analysis but it has also several weaknesses that the authors might address:

  1. the introduction is too much focused on the previous work and does not present the state of the art of knowledge about the field. 

We have added in a number of additional references and updated information throughout the introduction

2. There are results that have not been discussed from the perspective of the existing literature (e.g., immunological response).

We agree this is an important angle from which to consider the paper, and have added in some comments throughout the discussion accordingly

3. How is different the baseline evaluation of the responders from the non-responders? Have you looked for some predictors of drop-out

We have examined for differences in neurological symptoms between the returning sample and the original cohort, finding none (this is included in the second paragraph of the results). As this supports that the current sample is comparable to the original sample we do not feel that any further analysis seeking to separate those who returned and those who didn’t should be necessary. We hope the reviewer understands this position. We have added in a line in the results to clarify this, and a section to the “Statistical Analysis” part of the Methods to better highlight that the comparison has taken place.

Round 2

Reviewer 2 Report

I think the authors have substantially improved the manuscript with a deeper introduction and with a discussion that includes also future prospective. I think the paper could be considered for publication.

Author Response

-